# Security Providing Leadership: A Job Resource to Prevent Employees’ Burnout

**DOI:** 10.3390/ijerph182312551

**Published:** 2021-11-28

**Authors:** Juan A. Moriano, Fernando Molero, Ana Laguía, Mario Mikulincer, Phillip R. Shaver

**Affiliations:** 1Department of Social and Organizational Psychology, Faculty of Psychology, Universidad Nacional de Educación a Distancia (UNED), 28040 Madrid, Spain; fmolero@psi.uned.es (F.M.); aglaguia@psi.uned.es (A.L.); 2Baruch Ivcher School of Psychology, Interdisciplinary Center (IDC), Herzliya 46150, Israel; mario@idc.ac.il; 3Department of Psychology, University of California, Davis, CA 95616, USA; prshaver@ucdavis.edu

**Keywords:** leadership, security provider, attachment theory, burnout, organizational climate, organizational dehumanization

## Abstract

Leadership styles in work contexts play a role in employees’ well-being, contributing to better health or, on the contrary, being a source of stress. In this study we propose that security providing leadership may be considered as a resource to prevent employees’ job burnout. First, we examine the relationship between employees’ perception of their leader’s degree of security in providing leadership and the employees’ degree of job-related burnout. Second, the underlying processes by which leaders as security providers exert their influence on burnout are analyzed with a focus on the mediating role of two variables: an organizational climate oriented to psychological safety and organizational dehumanization. A total of 655 Spanish employees (53.7% women) completed a paper-and-pencil self-report questionnaire. To recruit participants, we employed an exponential non-discriminative snowball sampling. Results, using Partial Least Squares Structural Equation Modeling (PLS-SEM) to test hypotheses, show that security providing leadership was related negatively to burnout. Furthermore, psychological safety climate and organizational dehumanization mediated the relationship between security providing leadership and burnout. These findings support the attachment approach to leadership and open new avenues for creating better organizational environments. Security-providing leaders, by supporting employees and treating them in a personalized way, can enhance the psychological safety climate and prevent organizational dehumanization and consequent job burnout.

## 1. Introduction

Leadership consists of guiding, motivating, and inspiring employees to achieve an organization’s objectives [1,2]. A leader must influence followers to want to do what is requested of them, and to internalize organizational goals as appropriate and valid. This has nothing to do with brute force or raw power. Certainly, either of these strategies can be used to shape others’ behavior, but under duress subordinates will perform the requested activities with little enthusiasm, will not achieve optimal results, and will tend to disappear as soon as possible. In contrast, effective leadership involves influencing others in such a way that they are encouraged to contribute voluntarily to the achievement of organizational goals [3].

The study of leadership has traditionally focused on analyzing the effects of a leader’s characteristics and behaviors on employees’ performance and satisfaction. Although there are many different approaches to leadership, it should be pointed out that the appearance of a new model of leadership does not eliminate previous models; many of them continue to coexist. Thus, at present, work continues to be published from the perspective of personality traits and leadership styles (e.g., transformational, ethical, servant or authentic leadership), and contingency theories continue to maintain some influence.

However, focusing only on aspects of leadership underlying improved individual or organizational performance has led to the neglect of other important consequences of organizational leadership. Managers and supervisors also play a vital role in the prevention of occupational hazards; they are responsible for establishing effective policies and practices that promote the safety, protection, and health of workers. Moreover, the way leadership is exercised can have beneficial (protective) or detrimental (risky) consequences for the well-being and health of workers. The literature (e.g., [4,5,6]) shows that good leadership protects employees’ health and reduces their levels of stress and burnout, whereas negligent or poor leaders are an important source of stress for their subordinates, and many employees report that the worst aspect of their job is their immediate boss [7].

The objective of the present research is to explore the influence of security providing leadership on employees’ job burnout. According to the Job Demands-Resources (JD-R) theory [8], positive forms of leadership (e.g., transformational leadership) can be considered a job resource for employees [9]. In the same vein, we propose that security providing leadership, a type of organizational job resource, can prevent burnout through two mediating factors. First, security providing leadership can contribute to a psychological safety climate, which is a supportive job resource that protects employees from burning out. Second, security providing leadership can reduce organizational dehumanization—a well-known risk factor for employees’ burnout.

### 1.1. Job Burnout

Burnout is a prolonged response to chronic emotional and interpersonal stressors in the workplace. This response is characterized by (a) emotional exhaustion –feelings of being depleted emotionally and physically, (b) cynicism– negative responses to the workplace that frequently lead to depersonalizing the customers or recipients of services, and (c) feelings of incompetence at work [10]. Burnout arises primarily due to employees experiencing heightened stress in the workplace. It affects both individuals (by diminishing physical and mental health) and organizations (by decreasing employees’ motivation and performance). As the JD-R theory stands, this work-related syndrome is associated with emotional and situational demands (e.g., high workload) along with a lack of job resources (e.g., low social support) to cope with these demands [11].

Previous studies have shown that leaders can help to prevent employees’ burnout. Particularly, leaders who have regular contact with their followers may pay individual attention to their personal needs and use one-on-one coaching and mentoring to reduce job strain [11]. Furthermore, Kaluza et al. [12] found that leaders’ health awareness was positively related with their health-promoting leadership behaviors, which eventually went along with decreasing employees’ emotional exhaustion. In this research, we will analyze the extent to which subordinates’ perception of their leader as a security provider constitutes a job resource that can reduce their degree of burnout.

### 1.2. Security Providing Leadership

The conception of the leader as a security provider or a secure base for their subordinates is based on an attachment perspective on leadership. According to attachment theory [13,14,15,16], a security-providing figure serves five purposes: (1) secure base—he or she allows an adult relationship partner to pursue nonattachment goals in a safe environment; (2) safe haven—he or she reliably provides protection, comfort, support, and relief in times of need; (3) he or she is a target for proximity seeking when help is needed; (4) emotional bond—people tend to feel attached or connected to a person who they know cares for them; (5) separation distress—people react with intense distress to actual or potential unwanted separations from or losses of an attachment figure. A leader may fulfill all of these functions. For instance, effective leaders [17] are likely to be available, sensitive, and responsive to their followers’ needs; provide advice, guidance, and emotional and instrumental resources to group members; develop followers’ autonomy, initiative, and creativity; build followers’ sense of self-worth, competence, and mastery; and support their desire to take on new challenges and acquire new skills.

This attachment approach to leadership has a solid theoretical base and has been empirically validated in previous studies. Molero et al. [18] created a measurement instrument to assess the extent to which employees recognize their leaders as security-providing attachment figures. In a first study, they found that security providing leadership explained variance in employees’ satisfaction with their manager and perception of the manager’s efficacy above and beyond transformational leadership. In a second study, security providing leadership was positively associated with employees’ organizational identification, work engagement, and job satisfaction. In a third study, security providing leadership had a protective influence on job burnout. Specifically, employees’ perceptions of their leader as a security provider heightened their positive emotions and reduced their negative emotions at work, which in turn prevented emotional exhaustion and cynicism and sustained professional efficacy. Here, expecting to replicate these findings, we hypothesize the following:

**Hypothesis** **1** **(H1).**
*Security providing leadership will be negatively related to employees’ job burnout.*


### 1.3. Psychological Safety Climate

A psychological safety climate refers to “formal and informal organizational practices and procedures guiding and supporting open and trustful interactions within the work environment” [19]. Therefore, a work environment that includes a climate for psychological safety is one in which employees feel safe to be frank without being reprimanded or chastised.

According to Newman et al. [20], supportive job resources generate a psychological safety climate, conferring protection from resource loss, which in turn is related to negative individual outcomes such as stress and team conflict. One of the most important job resources is supportive leadership behaviors. Researchers have argued that when employees are supported by their leader, they will reciprocate with supportive behaviors themselves, creating a psychologically safe environment for the rest of their team [21]. Moreover, psychological safety climate requires leadership commitment to provide resources to employees, so they can effectively perform their jobs and ameliorate distress [22]. Thus, a psychological safety climate is affected by the extent to which leadership deals with the issues of psychological health and safety as well as organizational production goals [22]. In line with this reasoning, the following hypothesis is proposed:

**Hypothesis** **2** **(H2).**
*Security providing leadership will be positively related to an organizational climate of psychological safety.*


The lack of a psychological safety climate alerts employees to the threats or hazards (e.g., possible failure or rejection) associated with their work. Specifically, when employees perceive a lack of psychological safety, and instead must worry about how to act, their behavioral inhibition system (e.g., keeping silent) is likely to be engaged more frequently [15]. For example, employees may avoid reporting work overload and fatigue. Moreover, the lack of psychological safety could increase pressure to hide emotions rather than express them. When an employee suppresses ideas or concerns, this can create emotional dissonance between what the employee really believes (e.g., “this issue is important to share with my team leader”) and how they behave (withholding comments on the issue). This emotionally dissonant state has been strongly associated with burnout [23,24]. This leads to the following prediction:

**Hypothesis** **3** **(H3).**
*Psychological safety climate will be negatively related to employees’ job burnout.*


### 1.4. Organizational Dehumanization

Dehumanization is a grave and damaging form of social judgment. Regarding working environments, previous researchers have proposed the concept of organizational dehumanization, which means employees’ perception of being mechanistically dehumanized or objectified by their organization [25,26]. Employees in organizational settings often feel the extremely negative experience of being treated as interchangeable machines, numbers, or resources [27]. Organizational dehumanization is related to hierarchical relationships within an organization involving leadership [28] and power dynamics [29]. Research has shown that people with power have a decreased propensity to embrace others’ points of view, maintain a greater interpersonal distance [30], and increase the use of mechanisms of dehumanization [31]. Leadership style might be a major factor affecting organizational dehumanization. Previous studies have found that when a leader shows abusive behaviors, such as mocking employees, shouting at them, or depreciating them [32], employees feel dehumanized by their organization. Nevertheless, researches have just begun to explore the role of leadership in perceived organizational dehumanization [33,34]. One of the aims of this study is to analyse the positive role that security providing leadership could play in reducing organizational dehumanization, as well as its consequences (e.g., employees’ well-being). Therefore, we formulate the following hypothesis:

**Hypothesis** **4** **(H4).**
*Security providing leadership will be negatively related to organizational dehumanization.*


Organizational dehumanization has negative effects on employees’ well-being, attitudes toward the organization and work behaviors [35]. According to self-determination theory [36], psychological well-being depends on the satisfaction of basic psychological needs of autonomy, competence, and relatedness. However, organizational dehumanization reduces the possibility of meeting these needs and consequently has a negative effect on employees’ mental health, causing problems such as depression, anxiety, and stress-related disorders [27]. Prior research has shown that organizational dehumanization is associated with employees’ emotional exhaustion and psychosomatic strains [26,34,37]. To replicate those results, the following hypothesis is proposed:

**Hypothesis** **5** **(H5).**
*Organizational dehumanization will be positively related to employees’ job burnout.*


The full set of direct hypotheses is included within the model shown in Figure 1. In addition, we place special attention on the process of mediation. We propose that security providing leadership is related to employees’ job burnout through psychological safety climate and organizational dehumanization. The first path (H2–H3) predicts that security providing leadership will contribute to creating and maintaining a psychological safety climate and this, in turn, will decrease employees’ job burnout. The second path (H4–H5) proposes that security providing leadership will reduce organizational dehumanization, which in turn will be positively related with employees’ job burnout. This leads to the following mediation hypothesis:

**Hypothesis** **6** **(H6).**
*The impact of security providing leadership on employees’ job burnout is mediated by psychological safety climate and organizational dehumanization.*


## 2. Materials and Methods

### 2.1. Participants

The minimum sample-size comprised 652 participants based on G-power [38] for a small effect size in a regression model (one predictor, *f*^2^ = 0.02, α = 0.05, 95% power). The final sample included 655 Spanish employees, 53.7% of whom were women; their average age was 36.58 years (*SD* = 9.85), and their mean organizational tenure was 6.09 years (*SD* = 6.68). Most participants had a college degree (56.8%) or vocational training degree (20.2%). Participants belonged to 130 private (74.6%) and public (24.4%) organizations from several sectors: health (17.3%), education (14.1%), and administration (11.1%), among others. Most of the organizations were large (43.1%) or medium (29.8%) in size. The leader was male in most cases (62.7%).

### 2.2. Measures

After the participants had agreed to participate, they were given a questionnaire measuring the following variables:

Security providing leadership. To measure employees’ perceptions of their manager as a security-providing attachment figure, we used the 15-item Leader as Security Provider Scale (LSPS; [18]). The scale’s guidelines and all items were framed to focus participants on their direct manager or supervisor (e.g., “When I need help at work, I seek out my leader”). Participants were instructed to indicate the extent to which they agreed or disagreed with each item using a 7-point scale ranging from 0 (totally disagree) to 6 (totally agree). An overall score on this scale is calculated as the mean of the 15 item responses. Cronbach’s alpha and McDonald’s omega for this scale were high (α = 0.96 and ω = 0.96).

Psychological safety climate. This measure contained five items (e.g., “When someone in our organization makes a mistake, it is often held against them,” reverse scored) developed by Baer and Frese [19], which has been used in previous studies with Spanish employee samples [39]. Participants were asked to rate the extent to which they agreed with each item using a 7-point scale ranging from 0 (totally disagree) to 6 (totally agree). Cronbach’s alpha for this scale was 0.71 and McDonald’s omega was 0.72.

Organizational dehumanization. Participants were asked to state the degree to which they perceived that their organizations deemed them as resources, using 10 items (e.g., “My organization considers me to be a number”) from Caesens et al. [26], which has also been applied in previous studies conducted in Spain [33]. We computed an overall score by averaging the item responses. The answer alternatives ranged from 0 (completely disagree) to 6 (completely agree). Cronbach’s alpha and McDonald’s omega for this scale were high (α = 0.94 and ω = 0.94).

Burnout. We used the Spanish version of the MBI General Survey [40], adapted by Salanova et al. [41]. This 15-item scale measures three dimensions of burnout: emotional exhaustion (e.g., “I feel emotionally drained from my work”), cynicism (e.g., “I have become less enthusiastic about my work”), and professional efficacy (e.g., “I can effectively solve the problems that arise in my work”). All items are scored on a 7-point frequency rating scale ranging from 0 (never) to 6 (every day). For our analyses, the three dimension scores were averaged. Cronbach’s alpha was 0.65 and McDonald’s omega was 0.72.

Finally, participants were asked about demographic information (i.e., age and gender) and a variety of measures related to the characteristics of their institutions (e.g., type of institution and size of the organization).

### 2.3. Procedure

Participants completed a Spanish paper-and-pen questionnaire that comprised the scales described above, at the end of which a section requesting sociodemographic data was added. To recruit participants, we employed an exponential non-discriminative snowball sampling. First, we contacted Spanish university students in a master’s degree in occupational risk prevention program to request their involvement in this research with two conditions: (1) they were part of a workgroup with a least four members, even if they did not perform similar tasks or roles, and (2) the workgroup was managed by the same leader. These workers then asked their coworkers to participate in this study and gave them a packet including a document with instructions and assurance of the anonymity and confidentiality of their responses; the questionnaire; and an envelope to return the questionnaire to the coworker who handed it out to them. The questionnaire took no more than 15 or 20 min to complete.

### 2.4. Data Analysis

Descriptive statistics (means, standard deviations, correlations) were calculated using SPSS software (Statistical Package for the Social Sciences). Structural Equation Modeling (SEM) was used to test common method bias. Analysis was performed with IBM SPSS AMOS software using the original data matrix and following the maximum likelihood procedure [42]. Kline [43] suggests that the following indices should be reported: the model chi-square goodness-of-fit index, the Root Mean Square Error of Approximation (RMSEA), the comparative fit index (CFI) and the Standardized Root Mean Square Residual (SRMR). Values below 0.08 for RMSEA and SRMR indicate a good fit. For the CFI, values greater than 0.90 indicate a good fit, whereas values greater than 0.95 indicate superior fit [44].

Data were further analyzed using Partial Least Squares Structural Equation Modeling (PLS-SEM). This is a non-parametric technique, which is useful for analyzing complex mediation models and examining advanced options such as evaluating multiple mediators [45,46]. PLS has two strengths that make it well suited to this study. Like other SEM techniques (e.g., AMOS), PLS-SEM accounts for measurement error and provides more accurate estimates of mediation effects than regression analyses. Moreover, PLS-SEM was developed to avoid the necessity of large sample sizes and normal distributions of the data [47]. SmartPLS v3.0 software was used [48], selecting 5,000 samples for the bootstrapping procedure. PLS-SEM analyses follow a two-step approach given by Hair et al. [45]. Before hypotheses are tested (inner model), reliability and validity of the measures are assessed (outer measurement model).

## 3. Results

We took several steps, both procedural and statistical, to ensure that the risk of common method bias was minimized. First, the questionnaire was anonymous to reduce social desirability bias. Second, common method bias (CMB) was tested using Harman’s single factor test [49] because all data was self-reported from a single source (i.e., questionnaire survey). The total variance for a single factor was 38.86%, suggesting that CMB did not affect our data. Furthermore, the results revealed a poor fit of a one-factor model to the data: Chi-square (495) = 6706.11, *p* < 0.001; RMSEA = 0.14; CFI = 0.58; SRMR = 0.15. To confirm these results, further analyses were performed following the procedure suggested by Podsakoff et al. [49]. This approach consists of adding to the theoretical model a first-order factor with all the measures as indicators. The findings showed that the model fit improved, although none of the path coefficients corresponding to relationships between the indicators and the general method factor were significant. Based on these findings, CMB does not seem to be a problem in this study.

### 3.1. Outer Measurement Model

The outer measurement model examines the relationships between the observable indicators and the hypothesized latent variables. This analysis addresses the question of how well the identified measures predict or construct the latent variables. The hypothesized outer measurement model is a four-factor model consisting of LSPS, safety climate, organizational dehumanization, and burnout. The individual reliability of each indicator was assessed by analyzing the loads or simple correlations with their respective latent variable. All the relationships between the indicators and their constructs were significant (*p* < 0.001). In an acceptable analysis, the standardized outer loadings (λ) should be greater than 0.60 with a critical *t*-value over 1.96 for *p* < 0.05. [50]. The loadings of the 33 indicators on the four latent constructs in the present study were generally strong (λ > 0.60). However, the third item of safety climate (“In our company some employees are rejected for being different”) and the burnout dimension of professional efficacy did not reach the cut-off value (λ = 0.58 and λ = −0.42, respectively); thus, they were removed from the model. The reliability for the safety climate scale did not change substantially (α = 0.70 and ω = 0.70), but for the burnout scale it increased to Cronbach’s α of 0.72 (McDonald’s omega remained at 0.72). All Cronbach’s α coefficients reported in the measures section are equal to or higher than 0.70 (usually considered to indicate reasonable reliability).

Average Variance Extracted (AVE) values achieved the critical threshold of 0.50 (0.78 for burnout, 0.51 for safety climate, 0.64 for organizational dehumanization, and 0.64 for LSPS; [51]), indicating that the variance explained by each construct is larger than the variance due to measurement error. These results provide evidence of the internal consistency and convergent validity of the constructs. To assess the constructs’ discriminant validity, we examined the indicators’ cross-loadings (the loadings of all indicators on the corresponding construct were greater than any of their correlations, or cross-loadings, with other constructs) and applied the Fornell and Larcker [52] criterion by comparing the square root of the AVE values with the latent variable correlations. Table 1 provides the correlations between constructs and, on the diagonal, the square root of the AVE values. As can be seen, the square root of the AVE values of each construct is greater than the correlations with any of the other constructs. In addition, following Henseler, Ringle, and Sarstedt’s [53] recommendation concerning the assessment of the Heterotrait-Monotrait (HTMT) ratios, we ascertained that all of the HTMT ratios were below 0.85 (the maximum value obtained was HTMT dehumanization-burnout = 0.65). Bootstrapping was also applied to test whether the HTMT values were significantly different from 1; none of the 95% bias-corrected and accelerated (BCa) confidence intervals included the value 1. These three criteria converge to support the discriminant validity of the measurement models. Finally, because the maximum values of Variance Inflation Factors (VIF) were below the recommended value of 5 [45], there were no concerns about multicollinearity in the present study.

### 3.2. Descriptive Statistics and Correlations between Variables

Descriptive statistics and the correlation matrix are displayed in Table 1. The correlations give provisional support for the hypotheses. Security providing leadership and psychological safety climate are positively associated (*r* = 0.40, *p* < 0.01) and both are negatively related to organizational dehumanization (*r* = −0.42, *p* < 0.01; *r* = −0.54, *p* < 0.01) and burnout (*r* = −0.32, *p* < 0.01; *r* = −0.48, *p* < 0.01). As expected, organizational dehumanization is positively related to burnout (*r* = 0.55, *p* < 0.01). Among the control variables, leader’s gender (being a woman) is significantly related to security providing leadership (*r* = 0.15, *p* < 0.01), while tenure is negatively related to safety climate (*r* = −0.17, *p* < 0.05) and positively related to organizational dehumanization (*r* = 0.12, *p* < 0.01).

### 3.3. Hypotheses Testing

Figure 2 shows the relationships between all the variables proposed in the model, including the two mediators. The results indicate that security providing leadership is significantly related to psychological safety climate (ß = 0.51, *p* < 0.001) and organizational dehumanization (ß = −0.39, *p* < 0.001); thus, H2 and H4 are supported. Safety climate and organizational dehumanization are significantly related to burnout (ß = −0.24, *p* < 0.001 and ß = 0.41, *p* < 0.001, respectively), thus supporting H3 and H5. Further, the coefficients of determination of safety climate and dehumanization (*R*^2^ = 0.26 and *R*^2^ = 0.15, respectively) achieved the minimum value of 0.10 [54], indicating the predictive validity of the model. This overall model explains 35% of variance in burnout.

Table 2 reports the significance of the direct and indirect effects following the recommendations proposed by Cepeda-Carrión et al. [46]. Safety climate and organizational dehumanization mediate the link between security providing leadership and burnout. This is a full mediation given that the direct leadership—burnout association is no longer significant, and both the indirect effects and the total indirect effect are significant. Additionally, 84.9% of the variance explained by the total effect results from the two mediating paths. Given that this percentage is over 80%, full mediation is supported. Statistical analyses revealed no statistical differences between the two mediators in their contribution to the total effect (bootstrap 95% CI [−0.03, 0.11], bias corrected 95% CI [−0.03, 0.11]), such that neither mediator has a stronger effect than the other. The total effect of security providing leadership on burnout was −0.33 (*p* < 0.001).

The PLS path model’s predictive accuracy can also be assessed by calculating the Stone-Geisser´s predictive relevance, the *Q*^2^ value [47]. Applying the blindfolding procedure (*D* = 8) and the cross-validated redundancy approach [45], we found that burnout had the highest value (*Q*^2^ = 0.27), followed by safety climate (*Q*^2^ = 0.13) and, finally, dehumanization (*Q*^2^ = 0.10). As all values were greater than 0, results indicated that the model had predictive relevance for these constructs.

The last step in structural model evaluation is to assess the effect sizes (*f*^2^, Table 2, [47]). Security providing leadership had a large effect on safety climate (*f*^2^ = 0.35) and a medium effect on dehumanization (*f*^2^ = 0.18). Security providing leadership had no effect on burnout (*f*^2^ = 0.00), supporting the mediated relationship. Effect sizes on burnout were moderate for dehumanization (*f*^2^ = 0.20) and small in the case of climate (*f*^2^ = 0.06).

## 4. Discussion

Organizations can offer opportunities for employees’ growth and development. However, work environments may also be a source of considerable stress leading to burnout. Positive forms of leadership, such as security providing leadership, may be considered as organizational job resources that help to prevent employees’ burnout [9]. In this study, we tested a model according to which security providing leadership reduces employees’ job burnout by increasing psychological safety climate and reducing organizational dehumanization.

As expected, our results revealed a direct negative effect of security providing leadership on employees’ job burnout (H1), which is in line with previous studies based on other positive models of leadership [4,5,6]. Security providing leadership was also positively related to psychological safety climate (H2) and negatively related to organizational dehumanization (H4). Employees who perceive their leader as a source of security feel free to express their opinions without fear of being reprimanded or chastised. Consequently, employees do not feel that they must withhold concerns or ideas (i.e., keep silent). In addition, employees who perceive their leader as a source of security feel valued as human beings by their organization. This psychological safety climate and organizational re-humanization can provide employees with strength and fortitude to endure work-related demands and strains and sustain emotional well-being in the workplace.

Our results also revealed that psychological safety climate was negatively associated with employees’ burnout (H3). Previous studies have shown that lack of a psychological safety climate leads employees to hide emotions and feelings and creates emotional dissonance between what they really believe and how they behave, which is in turn associated with burnout [23,24]. Our findings also indicated that organizational dehumanization was positively associated with job burnout (H5). Research has shown that organizational dehumanization is associated with employees’ emotional exhaustion and psychosomatic strains [26,34,37]. Finally, the results also show that psychological safety climate and organizational dehumanization fully mediated the association between security providing leadership and burnout (H6). It appears that security providing leadership has an indirect beneficial effect on burnout by reducing organizational dehumanization and improving the psychological safety climate.

A clear implication of this study is that organizations can reduce and prevent job burnout by encouraging security providing leadership behaviors. To this end, employees should perceive that their leader is a source of a safe haven to which they can turn in case of problems or difficulties. Our findings reinforce previous studies on the perception of one’s leader as a provider of security [18]. Like security-enhancing attachment figures, effective leaders are reliably available, sensitive, and responsive to their employees’ needs; provide advice, guidance, and emotional resources to group members; build employees’ sense of self-worth, competence, and mastery; and affirm their ability to deal with challenges. In other words, leaders can be sensitive and responsive caregivers who provide their employees with a sense of security and a solid platform for autonomous growth. This approach to leadership inspires a sense of courage, hope, and dedication in employees, whereas an insecure approach to leadership generates anxiety, anger, and despair [55].

Moreover, security providing leaders can influence employees’ burnout by enhancing a psychological safety climate and diminishing organizational dehumanization. This may be especially relevant in the present crisis created by COVID-19, which arouses considerable fear and uncertainty. Hiring and training security providing leaders who, given our results, can provide a greater sense of safety and optimism, and should have positive effects on employees’ well-being, which in turn should contribute to employees’ health and strong participation. To promote a psychological safety climate and avoid burnout, managers and supervisors should be made aware that their role as attachment figures for many of their employees has an important influence on the employees’ sense of safety and recognition as human beings and can reduce the occurrence and costs of burnout.

Of course, the study has some limitations. First, self-report measures may contribute to potential sources of bias such as social desirability response set and lead to some inflation of the observed relationship between the measured constructs [49]. A small limitation is that we did not include the efficacy subscale of burnout, which was the least correlated with the other subscales and plays a minor role in the burnout syndrome [56]. Second, the cross-sectional research design does not allow confident causal conclusions. Thus, additional methodologies will be required to fully test our hypotheses, including multilevel analyses to discern the individual and group level of influence of leader behaviors related to security provision. However, our design provides significant insights and will be valuable as a first step in examining phenomena of interest longitudinally and experimentally (e.g., with manager training interventions). Third, burnout was the only outcome variable examined in this study. It would be interesting to analyze other outcomes. Performance and workgroup effectiveness would be especially interesting to study in future research to assess the organizational outcomes of security providing leadership and the mediating role of psychological safety climate and organizational dehumanization. Finally, the organizational context must be considered regarding the effects of the leader as a security provider. Although leaders who provide security for their employees are likely to always be beneficial, their relevance may be greater in high stress organizational settings, as for example in a military mission or in security and emergency teams (firefighters).

## 5. Conclusions

Managers and supervisors play an important role in subordinates’ well-being. In this research we have focused on a leadership style especially suited to reduce subordinates’ stress and job burnout: security providing leadership. The model proposed and confirmed here indicates that security providing leadership influences burnout by enhancing organizational climate for psychological safety and reducing employee’s organizational dehumanization. More research, with more complex designs, is needed to explore other variables and organizational characteristics related to security providing leadership and its effects.

## Figures and Tables

**Figure 1 ijerph-18-12551-f001:**
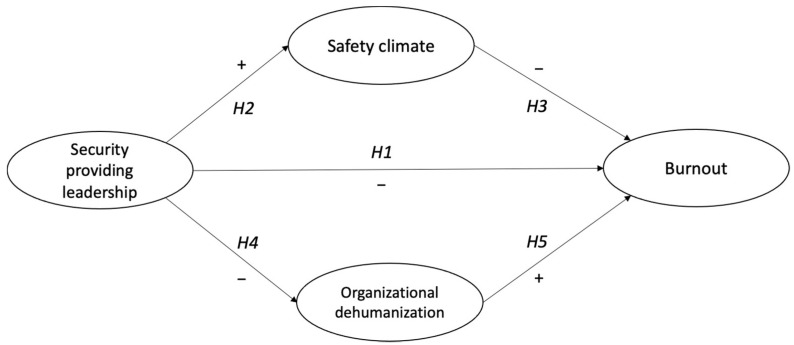
Theoretical model and hypotheses.

**Figure 2 ijerph-18-12551-f002:**
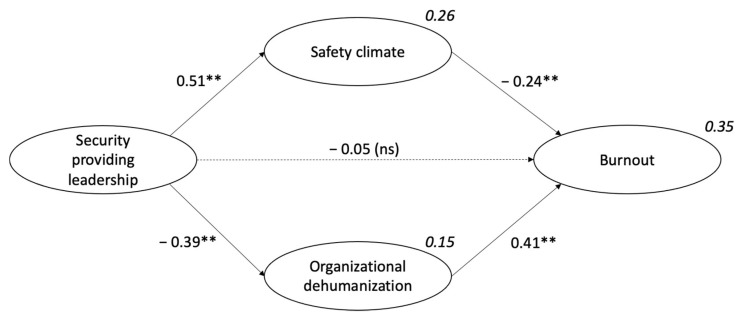
Standardized estimations for the full model. ** *p* < 0.01. Dotted lines show non-significant paths.

**Table 1 ijerph-18-12551-t001:** Means, standard deviations, correlations, and discriminant validity.

Constructs	Mean	SD	1	2	3	4	5	6
1. Leader’s gender (1 = female)	0.37	0.48	-					
2. Tenure	6.09	6.53	−0.03	-				
3. Security providing leadership	3.23	1.43	0.15 **	−0.03	0.80			
4. Safety climate	4.04	1.19	−0.01	0.17 **	0.40 **	0.72		
5. Organizational dehumanization	3.25	1.35	0.07	0.12 *	−0.42 **	−0.55 **	0.80	
6. Burnout	2.48	1.34	0.01	0.05	−0.32 **	−0.48 **	0.55 **	0.89

* *p* < 0.05, ** *p* < 0.01. √AVE estimates for latent variables are presented on the diagonal (based on PLS measurement models).

**Table 2 ijerph-18-12551-t002:** Mediating effects tests.

	Coefficient	Bootstrap 90% CI			
Direct effects		Percentile	BC		*f* ^2^
H1: Leadership—Burnout	−0.05	[−0.11, 0.02]	[−0.12, 0.02]		0.00
H2: Leadership—Climate	0.51 ^sig^	[0.46, 0.56]	[0.46, 0.56]		0.35
H4: Leadership—Dehumanization	−0.39 ^sig^	[−0.45, −0.33]	[−0.45, −0.34]		0.18
H3: Climate—Burnout	−0.24 ^sig^	[−0.30, −0.17]	[−0.31, −0.17]		0.06
H5: Dehumanization—Burnout	0.41 ^sig^	[0.35, 0.47]	[0.35, 0.47]		0.20
Indirect effects	Point estimate	Percentile	BC	VAF	
H2 × H3	−0.12 ^sig^	[−0.16, −0.09]	[−0.16, −0.09]	36.5%	
H4 × H5	−0.16 ^sig^	[−0.20, −0.13]	[−0.20, −0.13]	48.4%	
Total indirect effects	−0.28 ^sig^	[−0.33, −0.24]	[−0.33, −0.24]	84.9%	

Note. ^sig^: significant, CI: Confidence interval, BC: bias corrected, VAF: variance accounted for, *f*^2^: effect size.

## Data Availability

Data are available on request to the corresponding authors.

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
