# Peer review of "Security Providing Leadership: A Job Resource to Prevent Employees’ Burnout"

_ijerph, 2021, doi:10.3390/ijerph182312551_

Round 1

Reviewer 1 Report

This research paper seeks to understand how security providing leadership acts a mechanism to ameliorate employee’s job burnout. The author’s attempt to explain and provide insights through a mediated role of psychological safety climate and organizational dehumanization. The theoretical positioning of the construct to examine this phenomena are well-grounded in prior literature and theoretical underpinnings. 

There are several flaws in this paper that will require attention.  The writing needs to sharpen to reflect accuracy and a grasp of the English language. For example, in Introduction, section 1.1, second para. “security provider ‘rise’ should be ‘raise”.   In section, 1.2, para 3, “with such states being strongly associated with burnout does not make sense. The paper needs to be edited by someone who has mastered the English language.

I question the legitimacy of the exponential non-discriminative snowball sampling approach that starts from master’s degree students who are in occupational risk prevention that results the organizations across the sectors listed.  It would be important to know if the survey was distributed in Spanish or English language.

The proposed mediated for H5 is not argued to support mediation analysis.  The author’s propose that security providing leadership is “related” does not infer or suggest mediation.  Mediation seeks to identify and explain the mechanism or process that underlies an observed relationship between an independent variable and a dependent variable via the inclusion of a third hypothetical variable.  When observe ring a related relationship among constructs describes interaction or moderation effects. The justification for mediation needs to be rewritten.

The χ2 notation should be written out as chi-square. In the descriptive statistics and correlations between variables are not included in the Table 1.  For example, tenure is negatively related to safety climate and positively associate to organizational dehumanization.  In table 2, M1, the R2 is extremely week.  I would recommend using SmartPLS to perform the SEM analysis.  Amos is a package for estimating factor-based models. The composite-based approach to SEM uses weighted composites to represent unobserved conceptual variables. SmartPLS is one package for estimating composite-based models. The indirect effects (mediated) would provide a more robust results.  Moreover, very few researchers are using AMOS at this time.

There are several citations that are greater than 24 years old.  This references need to be updated and replaced with newer citations.

Overall, this paper needs significant improvements to become a publishable research article and a contribution to the literature.

Author Response

First, we would like to thank the reviewer for his thoughtful comments and suggestions towards improving our manuscript.

Our detailed, point-by-point responses to the reviewer comments are given below, whereas the corresponding revisions are marked in blue colored text in the manuscript file (version R1).

There are several flaws in this paper that will require attention.  The writing needs to sharpen to reflect accuracy and a grasp of the English language. For example, in Introduction, section 1.1, second para. “security provider ‘rise’ should be ‘raise”.   In section, 1.2, para 3, “with such states being strongly associated with burnout does not make sense. The paper needs to be edited by someone who has mastered the English language.

Thank you very much for your observations regarding the English language. We have corrected the errors you pointed out and the article has been proofread by one of the authors who is a native English speaker.

I question the legitimacy of the exponential non-discriminative snowball sampling approach that starts from master’s degree students who are in occupational risk prevention that results the organizations across the sectors listed.  It would be important to know if the survey was distributed in Spanish or English language.

We are aware that this form of sampling is not the best. However, our master's students are often professionals with work experience and allow us to access employee samples that would be difficult to obtain otherwise. In addition, all these students received clear instructions, and any questions they had were answered by the research team. In the Procedure subsection of the Method section, we have indicated that the questionnaire was distributed in Spanish.

The proposed mediated for H5 is not argued to support mediation analysis.  The author’s propose that security providing leadership is “related” does not infer or suggest mediation.  Mediation seeks to identify and explain the mechanism or process that underlies an observed relationship between an independent variable and a dependent variable via the inclusion of a third hypothetical variable.  When observe ring a related relationship among constructs describes interaction or moderation effects. The justification for mediation needs to be rewritten.

Hypothesis 5 refers to the direct relationship between organizational dehumanization and burnout. It does not involve mediation. Mediation is explained in the subsequent paragraph and in Figure 1.

The χ2 notation should be written out as chi-square. In the descriptive statistics and correlations between variables are not included in the Table 1.  For example, tenure is negatively related to safety climate and positively associate to organizational dehumanization.  In table 2, M1, the R2is extremely week.  I would recommend using SmartPLS to perform the SEM analysis.  Amos is a package for estimating factor-based models. The composite-based approach to SEM uses weighted composites to represent unobserved conceptual variables. SmartPLS is one package for estimating composite-based models. The indirect effects (mediated) would provide a more robust results.  Moreover, very few researchers are using AMOS at this time.

We have dealt with all of the issues you mentioned. In addition, we have followed your recommendation and used PLS-SEM with SmartPLS instead of AMOS. Therefore, data have been reanalyzed and the results section has been largely re-written to include the new PLS-SEM analyses. We believe this new analysis improved the quality of the manuscript.

There are several citations that are greater than 24 years old.  This references need to be updated and replaced with newer citations.

The majority of the old citations are related to attachment theory. We consider it important to mention the origins of this theory because it provides the conceptual framework for understanding the meaning and implications of security providing leadership. However, we have added some recent references concerning the association between leadership and burnout and about the Job Demands-Resources Theory.

Overall, this paper needs significant improvements to become a publishable research article and a contribution to the literature.

We believe that our revisions have improved the manuscript and we hope that it is ready for publications. Thank you again for your contributions.

Reviewer 2 Report

Dear authors,
I enjoyed reading your article, which is methodologically very well described. I have noticed a few small changes that I think could make the work even more interesting. Namely:
1) Although the methodological part is very complete and punctual, I noticed a lot of conciseness in the introductory theoretical part related to leadership. I suggest expanding on this: for each description of the constructs insert first a general definition, talk about the preeminent approaches on the domain (of traits, situational, behavioural) and then focus on the specific variable of your choice;
2) I recommend including a theoretical model that can act as a context to justify the hypotheses, such as the Job Demands-Job Resources model (Bakker, A. B., & Demerouti, E. (2017). Job demands-resources theory: taking stock and looking forward. Journal of occupational health psychology, 22(3), 273 - see Lord, F., Cortese, C. G., Parisi, S., Russo, V., Zito, M., & Ingusci, E. (2020). Job crafting and well-being at work: an exploratory analysis during health emergency period. Occupational Medicine, 111(6), 478-492. & Ingusci, E., Signore, F., Giancaspro, M. L., Manuti, A., Molino, M., Russo, V., ... & Cortese, C. G. (2021). Workload, Techno Overload, and Behavioral Stress During COVID-19 Emergency: The Role of Job Crafting in Remote Workers. Frontiers in Psychology, 12, 1141. for applications). 
3) Insert skewness and kurtosis measures to justify the use of parametric SEMs;
4) Alpha values greater than 0.95 could be a problem since they might be an indication of redundancy (Hulin, C., Netemeyer, R., & Cudeck, R. (2001). Can a reliability coefficient be too high?. Journal of Consumer Psychology, 55-58). The opposite is true for alpha = 0.65. Possibly include references in the literature that can confirm that these values are either too low or too high. I suggest using other indices, such as McDonald's omega or rho;
5) The representation of the structural relationship between safety climate and organisational dehumanisation is missing in Figure 1. Is this an oversight?
6) To define what kind of mediation was found consult Carrión, G. C., Nitzl, C., & Roldán, J. L. (2017). Mediation analyses in partial least squares structural equation modeling: Guidelines and empirical examples. In Partial least squares path modeling (pp. 173-195). Springer, Cham.

Author Response

First, we would like to thank the reviewer for his thoughtful comments and suggestions towards improving our manuscript.

Our detailed, point-by-point responses to the reviewer comments are given below, whereas the corresponding revisions are marked in blue colored text in the manuscript file (version R1).

1) Although the methodological part is very complete and punctual, I noticed a lot of conciseness in the introductory theoretical part related to leadership. I suggest expanding on this: for each description of the constructs insert first a general definition, talk about the preeminent approaches on the domain (of traits, situational, behavioural) and then focus on the specific variable of your choice;

Following the reviewer’s suggestions, we have inserted in the beginning of the manuscript some definitions of leadership before describing the construct of security providing leadership.

2) I recommend including a theoretical model that can act as a context to justify the hypotheses, such as the Job Demands-Job Resources model (Bakker, A. B., & Demerouti, E. (2017). Job demands-resources theory: taking stock and looking forward. Journal of occupational health psychology, 22(3), 273 - see Lord, F., Cortese, C. G., Parisi, S., Russo, V., Zito, M., & Ingusci, E. (2020). Job crafting and well-being at work: an exploratory analysis during health emergency period. Occupational Medicine, 111(6), 478-492. & Ingusci, E., Signore, F., Giancaspro, M. L., Manuti, A., Molino, M., Russo, V., ... & Cortese, C. G. (2021). Workload, Techno Overload, and Behavioral Stress During COVID-19 Emergency: The Role of Job Crafting in Remote Workers. Frontiers in Psychology, 12, 1141. for applications). 

Following the reviewer’s suggestion, we extended the presentation of the Job Demands-Job Resources Theory and its implications for the current findings in the introduction and discussion.

3) Insert skewness and kurtosis measures to justify the use of parametric SEMs;

Following Reviewer 1”s recommendations, we used PLS-SEM for data analysis. Therefore, it is no longer necessary to insert skewness and kurtosis measures. As expected, the results are very similar to those obtained before.

4) Alpha values greater than 0.95 could be a problem since they might be an indication of redundancy (Hulin, C., Netemeyer, R., & Cudeck, R. (2001). Can a reliability coefficient be too high?. Journal of Consumer Psychology, 55-58). The opposite is true for alpha = 0.65. Possibly include references in the literature that can confirm that these values are either too low or too high. I suggest using other indices, such as McDonald's omega or rho;

Thank you for recommending the inclusion of other reliability indexes. We have opted for the McDonald omega coefficient, which, unlike the alpha coefficient, works with factor loadings (Gerbing & Anderson, 1988), makes the calculations more stable (Timmerman, 2005), and reflects the true level of reliability. Moreover, it does not depend on the number of items (McDonald, 1999).

5) The representation of the structural relationship between safety climate and organisational dehumanisation is missing in Figure 1. Is this an oversight?

There is not a hypothesized association between safety climate and organizational dehumanization, and it is not the subject of the current study. The former Figure 2, which may have created some confusion, has now been corrected.

6) To define what kind of mediation was found consult Carrión, G. C., Nitzl, C., & Roldán, J. L. (2017). Mediation analyses in partial least squares structural equation modeling: Guidelines and empirical examples. In Partial least squares path modeling (pp. 173-195). Springer, Cham.

As mentioned above, we have now reanalyzed the data with PLS-SEM and followed all of the suggestions of the authors the reviewer mentioned (see Table 2). Thank you very much for this reference.

Reviewer 3 Report

Dear Sir or Madame,

first of all in many parts of the article it is

really difficult to understand what authors

would like to write about / there is a strong

need of language corrections. 
Title - I suggest to make it shorter and more

attractive as theme of the article is very

interesting.

Abstract: section with sample and

methods description should be added.

Introduction: first chapter is very hard to

understand. I like the way you present the

hypothesis but I thing research questions should be also

added. There is a lot in the introduction about

leadership’s style but I recommend to add

more about work burnout.

Results: I think it should be improved. More

data could be presented in form of tables

or figures - the clear description of obtained

results should be also written. I would

suggest to shorten the research procedure description.

The information about bioethical

commission’s approval is also needed.

Discussion: should be much improved -

logical flow is missing, link to the results is

missing, more international research

examples are needed - there have been a lot

of studies on the leadership and work

burnout carried out - those should be

presented in the discussion part.

Conclusions: should be shorter, more

clear and more precise.

In my opinion, the theme of the

paper is very interesting and important for both

employees and employers, the research seems

to be done professionally that is why I

suggest to prepare the paper better and

resubmit it when it will be improved. 

Author Response

First, we would like to thank the reviewer for his thoughtful comments and suggestions towards improving our manuscript.

Our detailed, point-by-point responses to the reviewer comments are given below, whereas the corresponding revisions are marked in blue colored text in the manuscript file (version R1).

Title - I suggest to make it shorter and more attractive as theme of the article is very interesting.

Thank you for your comment. We have changed the title of the article to the following: “Security Providing Leadership: A Job Resource to Prevent Employees’ Burnout”.

Abstract: section with sample and methods description should be added.

We have added the relevant information about the sample and the method in the Abstract.

Introduction: first chapter is very hard to understand. I like the way you present the hypothesis but I thing research questions should be also added. There is a lot in the introduction about leadership’s style but I recommend to add more about work burnout.`

Following your recommendations and those of other reviewers, the introduction has been expanded to include Job Demands-Job Resources Theory. In addition, we have added a new section (1.1.) on job burnout.

Results: I think it should be improved. More data could be presented in form of tables or figures - the clear description of obtained results should be also written. I would suggest to shorten the research procedure description. The information about bioethical commission’s approval is also needed.

Following Reviewer 1’s recommendation, we have reanalyzed the data with SmartPLS instead of AMOS. Therefore, the results section has been largely re-written to include the new PLS-SEM analyses.

According to the regulations of our university, the approval of the Ethics Committee is not required in the following cases: 

  1. If the research does not affect the fundamental rights (life, physical/psychic integrity, health, freedom/autonomy in any of its manifestations, personal dignity, etc.) of the subjects involved (on whom the research is based).
  2. If only non-identifying personal data are used.
  3. In teaching projects that only use data to which they have access as teachers.

Our research only requested participants to complete an anonymous questionnaire. Therefore, a report from the Ethics Committee was not required. Participants were provided with a research information sheet and their informed consent was collected.

Discussion: should be much improved - logical flow is missing, link to the results is missing, more international research examples are needed - there have been a lot of studies on the leadership and work burnout carried out - those should be presented in the discussion part.

Thank you for your suggestions. We have rewritten the discussion trying to make it clearer and including new references.

Conclusions: should be shorter, more clear and more precise.

We have shortened the conclusions trying to make it clearer and more precise

In my opinion, the theme of the paper is very interesting and important for both employees and employers, the research seems to be done professionally that is why I suggest to prepare the paper better and resubmit it when it will be improved. 

Thank you for your positive feedback.

Round 2

Reviewer 1 Report

The readability is tremendously improved and shows a command of the English language.  The H5 hypothesis is still not written to capture a mediating relationship. As mentioned previously, stating constructs are "related to" is not how mediation hypotheses are written. Mediation hypotheses, for example, are written like this, The impact of regional conditions on entrepreneurial intentions is mediated by attitude, subjective norm and perceived behavioral control.

The Fornell & Larcker results from SmartPLS need to be reported.  This standard reporting for any journal (Hair, Risher, Sarstedt, & Ringle, 2018). SmartPLS provides the indirect effect in the results of bootstrapping. Also the direct effect can be reported directed from there. Thus, no need to add mediators to the model to test the structural model.  All direct and indirect paths are processed simultaneously (Hair, Sarstedt, Ringle, & Gudergan, 2018).  The methods need to be written appropriately.  Please review the journal article, "When to use and how to report the results of PLS-SEM (Hair et al., 2018)

Hair, J. F., Sarstedt, M., Ringle, C. M., & Gudergan, S. P. (2018). Advanced issues in partial least squares structural equation modeling. SAGE.

Hair, J. F., Risher, J. J., Sarstedt, M., & Ringle, C. M. (2019). When to use and how to report the results of PLS-SEM. European business review, 31(1), 2-24.

Author Response

First, we would like to thank the reviewer for his or her thoughtful comments and suggestions.

Our detailed, point-by-point responses to the reviewer comments are given below, whereas the corresponding revisions are marked in purple colored text in the manuscript file (version R2).

R1: The readability is tremendously improved and shows a command of the English language. 

Thank you for your positive feedback. We really appreciate it.

R1: The H5 hypothesis is still not written to capture a mediating relationship. As mentioned previously, stating constructs are "related to" is not how mediation hypotheses are written. Mediation hypotheses, for example, are written like this, The impact of regional conditions on entrepreneurial intentions is mediated by attitude, subjective norm and perceived behavioral control.

Indeed, hypothesis 5 is not written to capture mediation, but to describe the direct relationship between organizational dehumanization and job burnout. Following the reviewer's suggestions, we have included a new hypothesis (H6) for mediation:
H6: The impact of security providing leadership on employees' job burnout is mediated by psychological safety climate and organizational dehumanization. 

R1: The Fornell & Larcker results from SmartPLS need to be reported.  This standard reporting for any journal (Hair, Risher, Sarstedt, & Ringle, 2018).

Following the reviewer's recommendations, we have included more information in Section 3.1. Outer measurement model and in Table 1.

To assess the constructs’ discriminant validity, we examined the indicators’ cross-loadings (loadings of all indicators on the corresponding construct were greater than any of their correlations, or cross-loadings, with other constructs) and applied the Fornell and Larcker [52] criterion by comparing the square root of the AVE values with the latent variable correlations. Table 1 provides the correlations between constructs and, on the diagonal, the square root of the AVE values. As can be seen, the square root of the AVE values of each construct is greater than the correlations with any of the other constructs.

R1: SmartPLS provides the indirect effect in the results of bootstrapping. Also the direct effect can be reported directed from there. Thus, no need to add mediators to the model to test the structural model.  All direct and indirect paths are processed simultaneously (Hair, Sarstedt, Ringle, & Gudergan, 2018).  

We have deleted two paragraphs from section 3.3. Hypotheses testing and added a new paragraph after Table 2.

The PLS path model´s predictive accuracy can also be assessed by calculating the Stone-Geisser´s predictive relevance, the Q2 value [47]. Applying the blindfolding procedure (D = 8) and the cross-validated redundancy approach [45], we found that burnout had the highest value (Q2 = .27), followed by safety climate (Q2 = .13) and, finally, dehumanization (Q2 = .10). As all values were greater than 0, results indicated that the model had predictive relevance for these constructs.

The last step in structural model evaluation is to assess the effect sizes (f2, Table 2, [47]). Security providing leadership had a large effect on safety climate (f2 = .35) and a medium effect on dehumanization (f2 = .18). Security providing leadership had no effect on burnout (f2 = .00), supporting the mediated relationship. Effect sizes on burnout were moderate for dehumanization (f2 = .20) and small in the case of climate (f2 = .06).

R1: The methods need to be written appropriately.  Please review the journal article, "When to use and how to report the results of PLS-SEM (Hair et al., 2018)

Hair, J. F., Sarstedt, M., Ringle, C. M., & Gudergan, S. P. (2018). Advanced issues in partial least squares structural equation modeling. SAGE.

Hair, J. F., Risher, J. J., Sarstedt, M., & Ringle, C. M. (2019). When to use and how to report the results of PLS-SEM. European business review, 31(1), 2-24.

Thank you for the references. In the article we have mainly relied on the following references:

  1. Hair, J. F.; Hult, G. T. M.; Ringle, C. M.; Sarstedt, M., A primer on Partial Least Squares Structural Equation Modeling (PLS-SEM). 2nd ed.; Sage: Los Angeles, CA, USA, 2017. 
  2. Cepeda-Carrión, G.; Nitzl, C.; Roldán, J. L., Mediation analyses in partial least squares structural equation modeling: Guidelines and empirical examples. In Partial least squares path modeling, Latan, H.; Noonan, R., Eds. Springer: Cham, 2017; pp 173–195. 
  3. Hair, J. F.; Risher, J. J.; Sarstedt, M.; Ringle, C. M., When to use and how to report the results of PLS-SEM. European Business Review 2019, 31, 2–24. 

Reviewer 3 Report

Dear Authors,

Thank you very much for your work on

improvement of the paper. I find it now much

more logical and coherent.

As I previously have written the issue of work

burnout became as serious issue for both

employees and employers so I think that the

more research on that subject the better

and I encourage you to carry on with

your analysis on this matter.

Best regards,

Author Response

Thank you very much for your words of appreciation and encouragement. We will certainly continue our research on employees’ job burnout and its relationship with leadership.